# S-glutathionylation proteome profiling reveals a crucial role of a thioredoxin-like protein in interspecies competition and cariogenecity of *Streptococcus mutans*

**Zhengyi Li**[1], **Chenzi Zhang**[1,2], **Cheng Li**[1,2], **Jiajia Zhou**[1,2], **Xin Xu**[1,2], **Xian Peng**[1]*, **Xuedong Zhou**[1,2]*

**1** State Key Laboratory of Oral Diseases, National Clinical Research Center for Oral Diseases, West China Hospital of Stomatology, Sichuan University, Chengdu, China, **2** State Key Laboratory of Oral Diseases, National Clinical Research Center for Oral Diseases, Department of Cariology and Endodontics, West China Hospital of Stomatology, Sichuan University, Chengdu, China

* pengx@scu.edu.cn (XP); zhouxd@scu.edu.cn (XZ)

**Data Availability Statement:** All proteomics raw data are available form PRIDE database (Accession number: PXD019564).

## Abstract

S-glutathionylation is an important post-translational modification (PTM) process that targets protein cysteine thiols by the addition of glutathione (GSH). This modification can prevent proteolysis caused by the excessive oxidation of protein cysteine residues under oxidative or nitrosative stress conditions. Recent studies have suggested that protein S-glutathionylation plays an essential role in the control of cell-signaling pathways by affecting the protein function in bacteria and even humans. In this study, we investigated the effects of S-glutathionylation on physiological regulation within *Streptococcus mutans*, the primary etiological agent of human dental caries. To determine the S-glutathionylated proteins in bacteria, the Cys reactive isobaric reagent iodoacetyl Tandem Mass Tag (iodoTMT) was used to label the S-glutathionylated Cys site, and an anti-TMT antibody-conjugated resin was used to enrich the modified peptides. Proteome profiling identified a total of 357 glutathionylated cysteine residues on 239 proteins. Functional enrichment analysis indicated that these S-glutathionylated proteins were involved in diverse important biological processes, such as pyruvate metabolism and glycolysis. Furthermore, we studied a thioredoxin-like protein (Tlp) to explore the effect of S-glutathionylation on interspecies competition between oral streptococcal biofilms. Through site mutagenesis, it was proved that glutathionylation on Cys41 residue of Tlp is crucial to protect *S. mutans* from oxidative stress and compete with *S. sanguinis* and *S. gordonii*. An addition rat caries model showed that the loss of S-glutathionylation attenuated the cariogenicity of *S. mutans*. Taken together, our study provides an insight into the S-glutathionylation of bacterial proteins and the regulation of oxidative stress resistance and interspecies competition.

**Funding:** The research performed in the Zhou lab was funded by the National Natural Science Foundation of China (81170959). The research performed in the Peng lab was funded by the National Natural Science Foundation of China (81700963) and Sichuan Science and Technology Program (2018JY0561). The funders had no role in study design, data collection and analysis, decision to publish, or preparation of the manuscript.

**Competing interests:** The authors have declared that no competing interests exist.

## Author summary

S-glutathionylation is involved in many physiological processes such as antioxidation, detoxification and cell proliferation. However, the proteomics research on S-glutathionylation has never been performed in gram-positive bacteria. The identification of and functional studies on glutathione reductase, glutathione synthesis dual-functional enzyme and glutathione-S-transferase in *Streptococcus mutans*, the main causative agent of dental caries, inspired the existence of complete S-glutathionylation system in this type of bacteria. It may impact various crucial physiological processes by regulating the function of modified proteins. Here, we explored the entire S-glutathionylation proteome through high-specific labeling and high-sensitivity mass spectrometry technologies. The results showed that about 11.7% of the total *S. mutans* proteins were S-glutathionylated, which indicated that S-glutathionylation plays a fundamental role in the regulation of cellular processes. Through functional analysis and domain prediction, we next uncovered a potential thioredoxin that contains two modified cysteines on its key active motif; S-glutathionylation of the sites affected the oxidative resistance, competition and cariogenecity of *S. mutans*. We anticipate that S-glutathionylation proteome studies could be performed on more refractory infectious diseases, revealing more mechanisms of pathogen infection and providing new methods to treat infectious diseases.

## Introduction

S-glutathionylation is the specific post-translational modification of protein cysteine thiols by the addition of GSH, which causes an increase in molecular mass and negative charge. This reversible modification not only protects cysteine from irreversible oxidation, but also causes significant structural and functional changes in the target protein, which leads to S-glutathionylation playing crucial roles in many physiological processes, such as antioxidation, detoxification, and the regulation of cell signal transduction and cell proliferation [1–3]. Although there have been some large-scale proteomics studies directed at identifying protein S-glutathionylation in mammalian and plant cells, only a few S-glutathionylation proteomics studies have been performed on bacteria, such as *Salmonella typhimurium* and *Synechocystis* sp. [4, 5].

The role of GSH in the regulation of various biological processes has been studied in a few gram-positive bacteria. For example, in *S. pneumoniae*, the glutathione reductase Gor and the GSH importer protein GshT were required for oxidative stress and metal ion resistance. A subsequent mouse pneumococcal infection model indicated that the colonization and invasion were attenuated after the *gshT* knockout [6]. In the intracellular pathogen *Listeria monocytogenes*, the GSH synthase gene *gshF* mutant was two-fold less virulent compared to the wild type (WT) in the mouse model and was sensitive to oxidative stress [7]. Thus, GSH might play important roles in regulating the virulence and host immune defense of gram-positive bacteria. However, the role of protein S-glutathionylation in gram-positive bacteria remains largely unknown because previous studies have suggested that most gram-positive bacteria use mycothiol (MSH; acetyl-Cys-GlcN-Myoinositol) and bacillithiol (BSH; Cys-GlcN-malate) as alternative redox buffers for protein S-thiolations [8–10].

*S. mutans* is considered the most cariogenic and prevalent species in active carious lesions. To compete against other oral bacteria that occupy the same ecological niche in dental plaque, *S. mutans* possesses unique adherence and metabolic attributes. It obtains energy by fermenting carbohydrates from the host diet and produces lactic acid as a byproduct. This byproduct

causes the pH to be maintained in the range of 4.5–5.5, resulting in the demineralization of enamel and subsequent formation of caries [11, 12]. In addition to constructing a low pH environment as a competitive strategy to inhibit other oral bacteria, *Streptococcus mutans* also uses its ability to resist the toxic effects of reactive oxygen species (ROS) derived from hostile bacteria, especially *Streptococcus sanguinis* and *Streptococcus gordonii*. They are early colonizers of dental plaque and produce hydrogen peroxide ($H_2O_2$) during aerobic metabolism at a concentrations ranging from 0.12–0.20 mM when grown in single-species planktonic cultures [13].

Competition among *S. mutans*, *S. sanguinis* and *S. gordonii* has already been reported and is considered to be prevalent in dental plaque biofilms [14–16]. *S. mutans* can synthesize a variety of bacteriocins, often called mutacins, which can inhibit the growth of *S. sanguinis* and *S. gordonii*. Moreover, to resist the $H_2O_2$ secreted by these two streptococci, *S. mutans* possesses various oxidative stress tolerance pathways, including the synthesis of reductases, such as superoxide dismutase (SOD), thiol peroxidase, alkyl-hydroperoxide reductase, NADH oxidase, and a Dps-like peroxide resistance protein [17]. In addition, studies on the GSH transport proteins GshT and cysteine ABC importer TcyBC of *S. mutans* suggested that GSH imported from the extracellular environment may also be crucial in preventing oxidative damage and regulating the protein function [18, 19]. A genomic study on a well-characterized cariogenic human isolate *S. mutans* UA159 identified a GSH synthesis dual-functional enzyme encoding gene *gshAB*, which encodes a protein with functional domains of glutamylcysteine synthetase and glutathione synthetase [20]. A later study of *gshAB* revealed that this gene is essential for the competitiveness and prevalence of *S. mutans* UA159 by detoxifying the $H_2O_2$ produced by *S. sanguinis* [21]. However, to the best of our knowledge, the protein S-glutathionylation of *S. mutans* has not yet been explored.

In this study, we combined high-specific labeling with high-sensitivity mass spectrometry technologies to identify the entire S-glutathionylation proteome of *S. mutans* UA159 [22, 23]. A total of 357 cysteine S-glutathionylation sites on 239 proteins were involved in diverse vital biological processes and metabolic pathways. We further analyzed the effects of Tlp S-glutathionylation on the virulence and competitiveness of UA159, and the results highlighted that Tlp might fictionize to modulate the dominance of *S. mutans* in dental biofilms, thus contributing to the management of dental caries.

## Materials and methods

### Ethics statement

All rat experiments in this study were performed in accordance with the protocols and procedures approved by the institutional Animal Care and Use Committee of West China Hospital of Stomatology, Sichuan University (approval number WCHSIRB-D-2019-194). The animal care and use protocol adhered to the Chinese National Laboratory Animal-Guidelines for Ethical Review of Animal Welfare.

### Bacterial strains and growth conditions

*S. mutans* UA159, *S. sanguinis* ATCC10556 and *S. gordonii* DL1 used in this study were obtained from the Oral Microbiome Bank of China [24]. Competent *E. coli* BL21 (DE3) and DH5α cells were purchased from Tsingke Corporation (Beijing, China) and grown in Luria-Bertani (LB) broth at 37˚C and 200 rpm. In-frame deletion of *gshAB* (SMU_267c), *gsT* (SMU_1296), *tlp* (SMU_1788c), and *tlp*-C41A site-direct mutation were generated via allelic exchange in *S. mutans* UA159 as described in previous studies [25, 26]. The primers used in this study are listed in **S8 Table**. All of the resulting gene mutations were verified via DNA sequencing. These strains were routinely grown at 37˚C under anaerobic (80% $N_2$, 10% $CO_2$,

10% $H_2$) or aerobic (5% $CO_2$, 95% air) conditions in brain heart infusion (BHI) broth (Difco, Sparks, MD, USA).

## Protein extraction

The **WT, AB, and T** groups were treated in the same way. Cell pellets were resuspended in lysis buffer (250 mM HEPES, 2% SDS, pH 7.0), and N-ethylmaleimide(NEM) was added to the lysis buffer to block the free thiols (the final concentration was 100 mM), The samples were then sonicated 12 times for 10 s each time followed by intervals of 30 s on ice for cooling. The remaining unbroken cells and debris were removed by centrifugation (12,000 × g at 4˚C for 10 min), and the supernatant was transferred to a new centrifuge tube and incubated for 30 min at 55˚C in darkness. Then, the protein was precipitated with six times the volume of pre-cooled acetone for 4 h at –20˚C. After centrifugation (12,000 × g at 4˚C for 10 min), the supernatant was discarded. The remaining precipitate was redissolved in the buffer (8 M urea, 250 mM HEPES, 0.1% SDS, pH 7.5) and transferred to an ultrafiltration tube (0.5 mL, 10 kDa; Millipore, Burlington, MA, USA) to replace the buffer with enzymatic reaction buffer (1 M urea, 25 mM HEPES, pH 7.5). Finally, the protein concentration was determined with a bicinchoninia acid (BCA) kit (Beyotime Biotechnology, JiangSu, China) according to the manufacturer's instructions.

## Reduction of S-glutathionylation sites

For each sample, 300 μg of protein was transferred to a new tube and diluted to 1 μg/μL by adding reaction buffer to 300 μL. Then, a final concentration of 2.5 μg/μL glutaredoxin 1M (Grx1M), 0.25 mM glutathione disulfide (GSSG), 1 mM reduced nicotinamide adnine dinucle-otide phosphate (NADPH) and 4 U/mL glutathione reductase (GR) were added to the reaction system. After reaction at 37˚C for 10min, the reaction was stopped on ice. Finally, the reaction buffer was replaced with labeling buffer by using ultrafiltration tubes, and the protein concentration was determined with a BCA kit as described above.

## Labeling the S-glutathionylated sites with iodoTMT

For labeling, 300 μg of reduced protein from each sample was transferred to new tubes and supplemented to the same volume by adding labeling buffer. The samples were labeled according to the manufacturer's instructions for the iodoTMT kit (ThermoFisher Scientific, San Jose, CA, USA). Briefly, the labeling reagents were dissolved in methyl alcohol and mixed with the protein and incubated at 37˚C for 1 h. Finally, excess reagents were removed by acetone precipitation. The **WT** group was labeled with 126 Da iodoTMT reagent, **AB** was labeled with 127 Da reagent, and **T** was labeled with 131 Da reagent.

## Trypsin digestion

The labeled protein pellet was resuspended in 100 mM ammonium hydrogen carbonate solution. Trypsin was added at a 1:50 (trypsin: protein) mass ratio and digested overnight at 37˚C. Then, the peptides were reduced with 5 mM dithiothreitol (DTT) for 30 min at 56˚C and alkylated with 11 mM iodoacetamide (IAM) for 15 min at 25˚C in darkness. Trypsin was then added at a 1:100 mass ratio for the second 4 h of digestion. The tryptic peptides were desalted using C18 ZipTips (Millipore) and dried by vacuum freezing.

## Anti-TMT antibody-conjugated resin-based enrichment

To enrich S-glutathionylated peptides, tryptic peptides were dissolved in NETN buffer (100 mM NaCl, 1 mM EDTA, 50 mM Tris-HCl, 0.5% NP-40, pH 8.0) and incubated with pre-washed anti-TMT antibody resins (Prod#90076, ThermoFisher Scientific) at 4°C overnight with gentle shaking in the dark. The resins were then washed four times with NETN buffer and twice with deionized water. The bound peptides were eluted from the resins three times with 0.1% trifluoroacetic acid. Finally, the enriched peptides were desalted and dried by vacuum freezing.

## Peptides analysis by high-performance liquid chromatography-tandem mass spectrometry (HPLC-MS/MS)

Peptide mixture analysis was performed using an Easy-nLC1000 nanoflow liquid chromatograph (ThermoFisher Scientific) as an HPLC system and a quadrupole-orbitrap hybrid mass spectrometer (Q Exactive Plus, ThermoFisher Scientific) as a mass analyzer. Peptides were dissolved in solvent A (0.1% formic acid) and directly loaded onto a home-made reversed-phase analytical column (15 cm length, 75 μm i.d.). The gradient was comprised of an increase from 10% to 23% solvent B (0.1% formic acid in 98% acetonitrile) in 38 min, 23% to 35% in 13 min and climbing to 80% in 4 min, and then holding at 80% for the last 5 min, all at a constant flow rate of 300 nL/min on an EASY-nLC 1000 UPLC system. The eluted peptides were subjected to a nanospray ionization (NSI) source followed by tandem mass spectrometry (MS/MS) in Q Exactive Plus coupled online to UPLC.

## Database search

Fragmentation data were then processed using the Maxquant search engine (v.1.5.2.8) against the *Streptococcus mutans* database containing 10523 sequences and concatenated with the reverse decoy database. Trypsin/P was specified as a cleavage enzyme allowing up to two missing cleavages and five modifications per peptide. The mass error for precursor ions was set to 20 ppm for the first search and 5 ppm for the main search, and the mass error for fragment ions was set as 0.02 Da. Peptides with lengths of at least seven amino acid residues were used for further analysis. S-glutathionylation on Cys was specified as fixed modification, and iodoTMT-6plex var (statement for isobaric label regent), acetylation of the protein N-terminal, and methionine oxidation were specified as variable modifications. Quantitation method was set to iodoTMT-6plex. The maximum false discovery rate (FDR) thresholds for proteins, peptides and modification sites were adjusted to < 1%.

## Protein annotation

The Gene Ontology (GO) annotation proteome was derived from the UniProt-GOA database (http://www.ebi.ac.uk/GOA/). The domain functional descriptions were annotated by InterProScan based on the protein sequence alignment method and the InterPro domain database (http://www.ebi.ac.uk/interpro/). The Kyoto Encyclopedia of Genes and Genomes (KEGG) database (https://www.kegg.jp/kegg/) was used to annotate the protein pathways. The prokaryotic organism subcellular localization prediction software CELLO was used to predict subcellular localization. The clusters of Orthologous Groups (COG) database (http://www.ncbi.nlm.nih.gov/COG) was used to align and classify the orthologs of the modified proteins.

## Functional enrichment

The biological processes, cellular components and molecular functions were categorized by GO annotation. The KEGG database was used to identify enriched pathways, whereas the InterPro database was used to enrich the protein domain. For all the above enrichment methods, the two-tailed Fisher's exact test was used to test the enrichment of the differentially expressed proteins against all identified proteins. Differences were considered significant when the corrected $P$-value < 0.05.

## Functional enrichment-based clustering

After functional enrichment, we collated all the categories and their P values, followed by filtering for those categories that were at least significantly enriched ($P$ < 0.05) in at least one of the clusters and transformed these $P$-value matrixes by the function X = –log10 ($P$-value). Then, these X values were Z-transformed for each functional category. These Z scores were finally clustered by one-way hierarchical clustering (Euclidean distance, average linkage clustering) in Genesis. The heatmap.2 function from the gplots R-package was used to visualize the cluster membership.

## Protein-protein interaction (PPI) network

All modified protein database accessions were searched against the STRING database version 10.5 (http://string-db.org) for the prediction of PPIs with a confidence score > 0.7 (high confidence). Only interactions between the proteins belonging to the searched data set were selected, thereby excluding external candidates. Interaction networks from STRING were visualized in Cytoscape 3.7.2 and clustered according to KEGG pathway enrichment.

## Protein purification and modification validation

To further validate the S-glutathionylated cystine modification results from proteomic analyses, targeted candidate proteins were cloned into the vector pET21a with a C-terminal 6 × His-tag and expressed in the *E. coli* BL21(DE3) strain, induced with 0.5mM isopropyl β-D-1-thio-galactopyranoside (IPTG) and grown at 37˚C for 4h. Then, the proteins were purified according to the manufacturer's instructions using the His-tag Protein Purification Kit (Beyotime Biotechnology). To modify the proteins, 2 μM purified proteins were incubated with 5 mM GSSG in 100 mM potassium phosphate buffer (pH 8.0) for 1 h at 37˚C, and an aliquot of the mixture was incubated with 80mM DTT for an additional 30 min at room temperature to validate the reversibility of this modification. After the treatments, the reagents were removed by dialysis. Then, non-reducing western blots were performed to identify the modification of these proteins. Mouse polyclonal anti-GSH antibody (#ab19534; Abcam, Cambridge, MA, USA) and mouse monoclonal anti-6 × His antibody (#66005-1-Ig, Proteintech, Hubei, China) were used to label the S-glutathionylated and 6 × His-tagged proteins, respectively. Samples were tested in triplicate and repeated at least three times.

## Bacterial competition on plates and in biofilms

For competition on plates, an overnight culture of either species was adjusted to OD$_{600nm}$ of ~ 0.5 in 50% BHI, 10 μL of which was inoculated on half-strength BHI plate as an early colonizer [15], 10 μL of the competing species at the same OD$_{600nm}$ was inoculated beside the early colonizer after overnight incubation, or both species were inoculated simultaneously. The plate was further incubated at 37˚C anaerobically overnight. All assays were performed in triplicate. For competition in biofilms, overnight cultures of *S. mutans*, *S. gordonii* and *S. sanguinis* were

diluted to $1 \times 10^5$ CFU/mL in fresh BHI supplemented with 1% sucrose and inoculated into slide chambers simultaneously. After 24 h of incubation, biofilms were fixed with 4% paraformaldehyde and subjected to species-specific fluorescence in situ hybridization (FISH), as described in a previous study [27]. All the groups were performed in triplicate and at least five randomly selected positions of each group were captured using an Olympus FLUPVIEW FV3000 confocal laser scanning microscope (Olympus Corp., Tokyo, Japan). The ratios of these species were analyzed with Image-Pro Plus 6.0 (Media Cybernetics. Inc., Silver Spring, MD, USA) and the 3D images were reconstructed using Imaris 7.0.0 (Bitplane, Zürich, Switzerland).

### Hydrogen peroxide challenge

The $H_2O_2$ sensitivity of *S. mutans* was evaluated as described in a previous study [28]. Briefly, mid-exponential-phase cells were harvested by centrifugation, resuspended in glycine buffer (0.1 M, pH 7.0), and exposed to 0.2% $H_2O_2$. After 30 min of exposure, catalase was added to a final concentration of 5 mg/ml to inactivate the $H_2O_2$. The survival rate was determined by plating in triplicate on BHI plates. All strains were tested in triplicate and repeated at least three times.

### Growth characteristics of the mutant strains

To construct the growth curve, ~$10^5$ CFU of mid-exponential-phase cells of each strain were inoculated into tubes containing 10 mL of BHI and incubated at 37°C under anaerobic or aerobic conditions. The $OD_{600nm}$ was determined at designated points using a SpectraMax M series Muti-Mode Microplate Reader iD3 (Molecular Devices). All strains were tested in triplicate and repeated at least three times.

### Rat model of dental caries

Specific-pathogen-free Spragur Dawley (SD) rats (Chengdu Dossy Experimental Animals Co., Ltd, China) were obtained at the age of three weeks and randomly assigned into three groups of five animals. All rats were provided with ampicillin (1 g/kg) in their drinking water for the first three days and then normal sterile water for another day to elute the antibiotic [29]. Then, the of *S. mutans* UA159 solution ($10^9$ CFU/mL, 0.3 mL) or the mutant strains were inoculated for three consecutive days. Simultaneously, cariogenic diet 2000 (*TrophicDiet*, Trophic Animal Feed, Suzhou, China) and sucrose-containing water (5% concentration) were provided. The rats were sacrificed after three weeks, and their maxillary and mandibular molars were obtained. All molars were dyed with ammonium purpurate (0.4% concentration) for 6 h, and then hemi-sectioned with a cutter. They were then examined with a stereomicroscope (Leica EZ4HD; Leica Microsystems AG, Heerbrugg, Switzerland) and scored utilizing the Keyes score method [30].

### Statistical analysis

Statistical analyses were performed using GraphPad Prism version 8.0 unless otherwise noted. $P < 0.05$ was considered significant.

## Results

### Identification of S-glutathionylated cysteine sites and proteins in *S. mutans* UA159

To obtain a systematic view of the protein S-glutathionylation proteome in *S. mutans*, S-glutathionylated peptides were labeled with iodoTMT isobaric reagents and enriched with resin

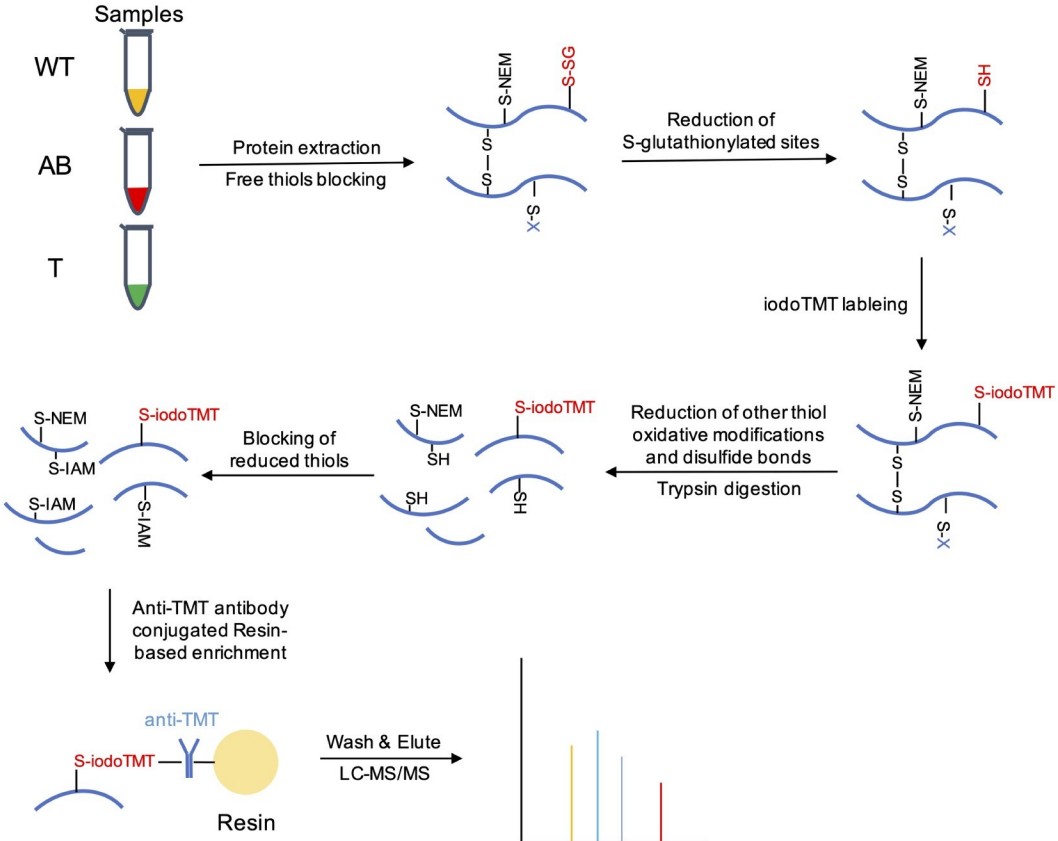

**Fig 1. Schematic overview of isotopic-labeling strategy to identify and quantify S-glutathionylated proteins.** NEM and IAM were used to alkylate free thiols. A cocktail of Grx, GR, GSH and NADPH was used to selectively reduce glutathionylated cysteines. 126 Da, 127 Da and 131 Da iodoTMT reagents were used to label groups WT, AB, and T, respectively. DTT was used to reduce disulfide bonds and other thiol oxidative modifications such as S-nitrosylation. Anti-TMT antibody-conjugated resins were used for specifically enrich the iodoTMT labeled peptides. The eluted peptides were subjected to LC-MS/MS analysis.

conjugated anti-TMT antibodies after GR reduction. The workflow of our proteomics study is shown in **Fig 1**. Then, the enriched peptides were identified using high-sensitivity MS/MS. A total of 357 S-glutathionylated sites of 316 peptides were identified on 239 proteins using a highly conservative threshold (FDR < 1%) (**S1 Table**). The near-zero distribution of mass error and high peptide score indicated that our method has a high accuracy for the identification of the S-glutathionylated peptides (**S1A Fig**). Approximately 97.45% of the total identified segment lengths were in the range of 7–21 segments, whereas 2.55% of total peptides exhibited lengths of 22–39 (**S1B Fig**). These results met the standards and requirements for MS data that are necessary for building a high-quality S-glutathionylation proteome of *S. mutans*. Moreover, the proportion of S-glutathionylated proteins among the total proteins was about 11.74% (239/2036); these proportions for in *Salmonella typhimurium* 14028 were 0.68% (38/55586) and 5.12% (383/7477) in *Synechocystis* sp. PCC6803 [4, 5], suggesting that protein S-glutathionylation plays an important role in the PTM of *S. mutans* UA159.

Most of the S-glutathionylated proteins contained between one to four modified sites, which accounted for 98.7% of the total PTM proteins, whereas the remaining 3 proteins had 5, 7 and 8 modified sites, respectively (**S1C Fig**). The protein Q8DUD3 (DQM59_RS05710), which is the DNA topoisomerase 1, had the highest number (eight) of S-glutathionylated sites.

Interestingly, all the modified sites of Q8DUD3 were in the region of 576–706, which is the last domain of C terminal, whereas other regions of this protein did not contain any cysteine residues, and the structure and function of this modified enriched region should be investigated.

## Subcellular localization prediction and functional classification of the S-glutathionylated proteins

To better characterize the role of S-glutathionylation in *S. mutans*, we first used WolfPsot software to predict the subcellular localization of the proteins corresponding to modification sites. Most of the S-glutathionylated proteins (215) were cytoplasmic, accounting for 89.96% of the total modified proteins (**Fig 2A** and **S2 Table**). The prevalence of S-glutathionylated proteins in the cytoplasm indicated that S-glutathionylation might be involved in various intracellular physiological pathways, such as biosynthesis, energy metabolism, biosynthesis, and substrate-binding. Other modified proteins residing in the membrane and extracellular space may be related to osmoprotectant transport, bacterial adhesion and resistance to fluctuating environments. For example, Q9KIJ3 (SMU_184) could be glutathionylated at C204. Q9KIJ3 is a metal ABC transporter substrate-binding membrane lipoprotein that plays essential roles in transporting metal ($Zn^{2+}$) and adherence to the extracellular matrix, which includes extracellular polysaccharides [31, 32]. Thus, these S-glutathionylation modifications in enzymatic and non-enzymatic proteins may indicate an involvement in multiple cellular physiological processes, including bacterial virulence and pathogenicity.

To further investigate the function of S-glutathionylation in regulating the cellular physiological processes of *S. mutans*, we conducted GO analysis to classify and enrich the molecular functions and biological processes involved in the modification of proteins. In the 'biological processes' category, the oxoacid metabolic process and carboxylic acid metabolic process were primarily enriched (adjusted Fisher's exact test p value < 0.001) (**Fig 2B** and **S3 Table**). In addition, other important organic molecules, involved in metabolic and synthetic processes, such as ribose phosphate, were also significantly enriched. Moreover, the molecular functions of the modified proteins were primarily enriched in ion and nucleotide binding, ligase activity, and oxidoreductase activity (**Fig 2C**). GO enrichment indicated that S-glutathionylated proteins were involved in energy metabolism, nucleotide transport and metabolism, redox homeostasis, translation processes, and substrate binding. Similar results were also summarized in the cluster of orthologous groups of protein (COG) functional classification (**S2 Fig** and **S4 Table**).

We used InterProScan to annotate the domain corresponding to modified sites, followed by enrichment analysis. The condensation domain, NAD(P)-binding domain, and AMP-dependent synthetase/ligase domain were the three most enriched domains (adjusted Fisher's exact test *P* < 0.001) (**Fig 2D** and **S5 Table**). The condensation domain has been reported in many multi-domain enzymes that synthesize peptide antibiotics. It catalyzes a condensation reaction in non-ribosomal peptide biosynthesis to form peptide bonds [33]. There were some putative peptide antibiotic synthetases containing the condensation domain in this study, including Q8DTJ6 (SMU_1341c), Q8DTJ7 (SMU_1340), Q8DTJ4 (SMU_1343c), and Q8DTJ5 (SMU_1342), that were glutathionylated at several cysteine sites. The primary modified AMP-dependent synthetase domain-containing proteins were peptide antibiotic synthetases. This indicated that the peptide antibiotic synthetases might exhibit a common regulatory mechanism via the S-glutathionylation of the condensation domain. Nineteen proteins, (including UDP-N-acetylmuramoylalanine-D-glutamate ligase, NADPH-dependent glutamate synthase (small subunit), redox-sensing transcriptional repressor Rex, and several

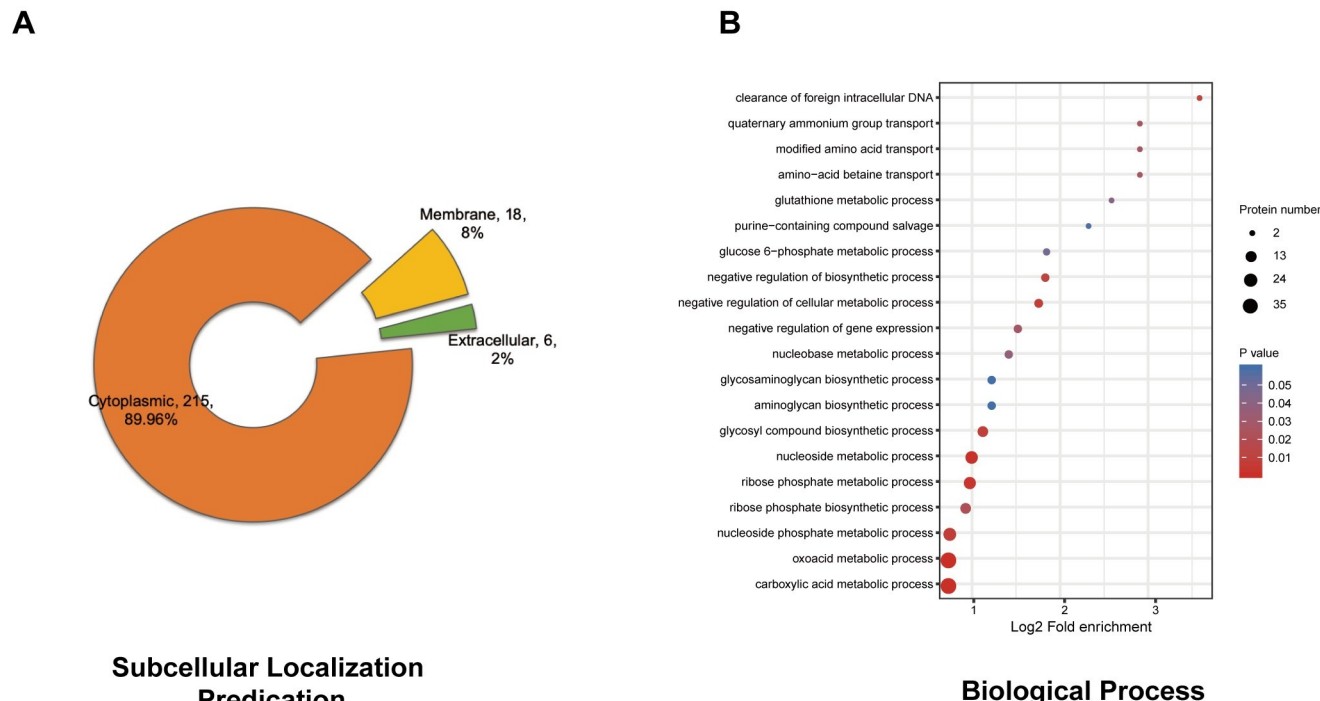

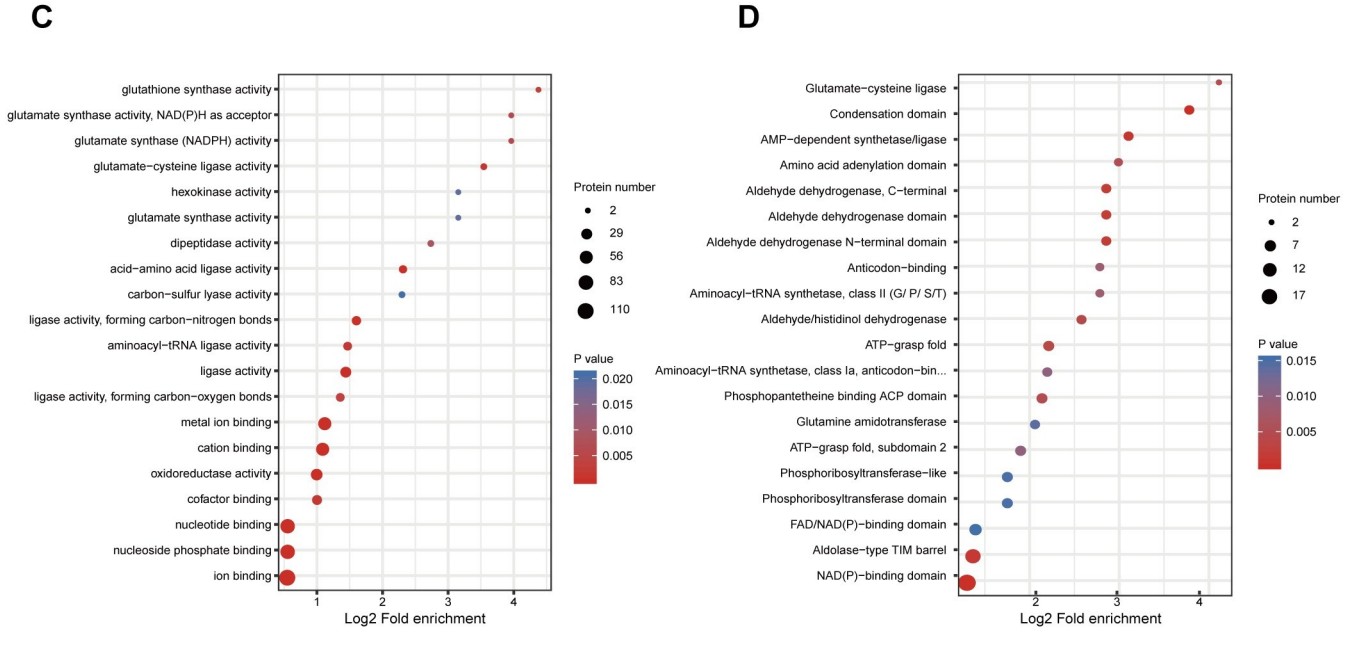

**Fig 2. Subcellular localization prediction and functional classification of identified S-glutathionylated proteins.** (A) Subcellular localization prediction of modified proteins. (B) The "Biological Process" category of GO-based enrichment analysis of modified proteins. (C) The "Molecular Function" category of GO-based enrichment analysis of modified proteins. (D) Enrichment analysis of domains related to modified proteins. For each category, a two-tailed Fisher's exact test was employed to test the enrichment of the identified modified proteins against all proteins in the species database. Corrected *P* values < 0.05 were considered significant.

important dehydrogenases) contained the modified NAD(P)-binding domain, which is involved in numerous physiological processes. In addition, there were also many important domains that could be S-glutathionylated, such as the aldehyde dehydrogenase N-terminal domain, ATP-grasp fold, and tRNA binding arm. Extensive research is needed to explain the function of S-glutathionylation in these domains and proteins.

## S-glutathionylation is involved in the crucial metabolic pathways of *S. mutans*

S-glutathionylation is involved in various metabolic pathways, both in prokaryotes and eukaryotes [34]. Thus, we assessed the enriched KEGG pathways of S-glutathionylated proteins in UA159. The modified proteins were primarily involved in the glycolysis/gluconeogenesis pathway (~5.4% of the total modified proteins), which is the central energy metabolic pathway of the cell (**Fig 3A**). The core enzymes of this pathway include fructose-1,6-biphosphate aldolase, ATP-dependent 6-phosphofructokinase, glucose kinase, and glyceraldehyde-3-phosphate dehydrogenase, all of which could be S-glutathionylated in their active domains (**S6 Table**). Moreover, seven modified proteins were enriched in the pentose phosphate

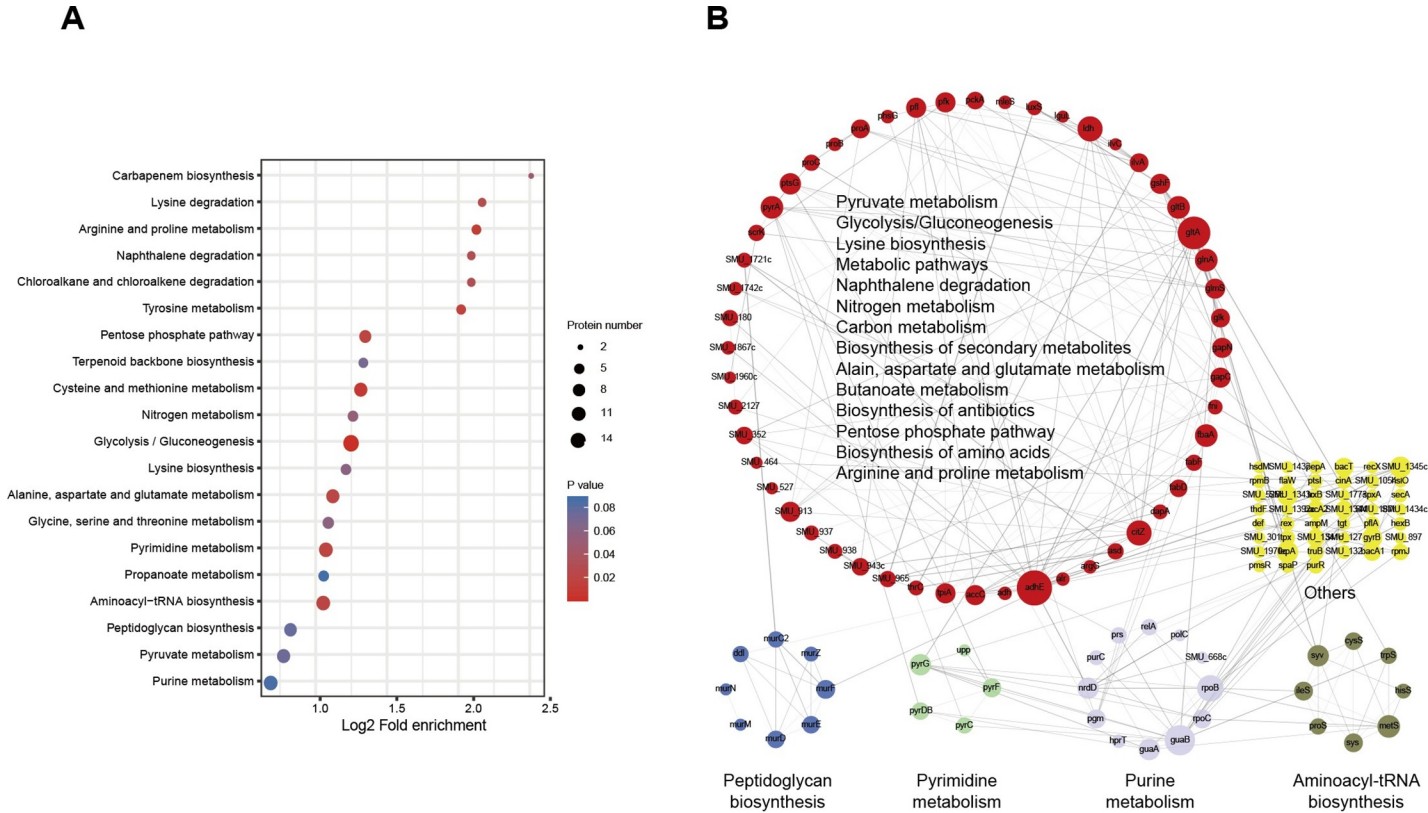

**KEGG Enrichment**                     **PPI Network**

**Fig 3. S-glutathionylation is involved in the crucial metabolic pathways of *S. mutans*** (A) KEGG pathway-based enrichment analysis of modified proteins. A two-tailed Fisher's exact test was employed to test the enrichment of the identified modified proteins against all proteins in the species database. Pathways with corrected $P < 0.05$ were considered significant. (B) STRING-based analysis of PPI networks of identified modified proteins. Different colors display various clusters according to KEGG pathway enrichment. The size of the node represents the value of the degree, and the width of the edge represents the EdgeBetweenness.

pathway, which is the principal source of NADPH and 5-phosphoribose. In addition, multiple amino acid metabolic pathways were also significantly enriched (adjusted Fisher's exact test $P < 0.05$), including cysteine, methionine, arginine, proline, tyrosine, pyrimidine, lysine, alanine, aspartate and glutamate. These results indicated that *S. mutans* might directly regulate core metabolic pathways through the S-glutathionylation of key enzymes.

The dynamic reversible modification of target proteins may affect the PPI, thus contributing to the regulation of metabolic processes [35]. The modified proteins were then evaluated for PPI networks and clustered into KEGG pathways using STRING (minimum required interaction score is 0.7). The representative KEGG pathways that were significantly enriched (corrected $P < 0.05$) as noted by PPI are shown in **Fig 3B.** The PPI pairs that could not be involved in the main network were excluded. Twelve purine and five pyrimidine metabolism-related proteins were S-glutathionylated and enriched. Of these, RpoB (SMU_1990) and RpoC (SMU_1989) are DNA-directed RNA polymerase subunits that play crucial roles in transcription and maintaining cellular viability; thus, they are targeted by antibiotics such as rifampicin [36]. Eight modified proteins were involved in the peptidoglycan biosynthesis pathway, which is involved in cell wall biogenesis. In addition, eight amino acid-tRNA ligases were involved in the aminoacyl-tRNA biosynthesis pathway, indicating the importance of S-glutathionylation modification in the regulation of translation. To highlight important metabolic pathway-related proteins that are involved in the PPI network, we listed some proteins corresponding to the core nodes and their functions (**Table 1**). As expected, proteins clustered into different metabolic pathways exhibited complicated PPI networks, suggesting that S-glutathionylation plays a critical role in PPIs, further participating in regulating multiple core metabolic pathways regulating, including the pentose phosphate pathway, glycolysis/gluconeogenesis, and pyruvate metabolism.

## S-glutathionylation of proteins is differentially modified after the deletion of *gshAB* and *gsT*

The *gshAB* gene encodes a bifunctional glutathione synthetase, which is involved in synthesizing GSH in *S. mutans*, while *gsT* encodes a Glutathione-S-Transferase (GST)-like protein, which conjugates GSH to a wide number of exogenous and endogenous hydrophobic electrophiles [20]. Both genes are important for protein S-glutathionylation of *S. mutans* and even modulate the competitive fitness of *S. mutans* by alleviating the oxidative stress exerted by the metabolites from other co-residents [21, 37]. However, the relationship between S-glutathionylation and the above modulating process has not been explored in *S. mutans*, and proteomics studies could be helpful in revealing the related mechanisms. Therefore, we performed the same proteomics method to identify and quantify the modified proteins together with the WT UA159 after the deletion of *gshAB* and *gsT*. Finally, 337 modified sites on 224 proteins were quantifiable (**S1 Table**). Differentially modification sites (proteins) are summarized in **Table 2** (filtered with a threshold value of modification fold change $> 1.2$ or $< 1/1.2$). Significantly differentially modified sites were clustered and displayed as a heatmap (**Fig 4A**). However, as the data shows, levels of S-glutathionylation were increased for part of the modified sites, indicating that there are other glutathionylation processes that independent of synthesis from GshAB and the enzymatic reaction of GST. For *gshAB* mutant, there might be other enzymes responsible for the complementary synthesis of GSH, however, there are no reports of other putative GSH-synthesizing enzymes. The non-enzymatic spontaneous S-glutathionylation process, which proceeds spontaneously especially when cellular oxidative stress increase, may play a predominant role after *gsT* deletion, [1].

**Table 1. Core proteins in PPI network based on KEGG clustering.**

| Protein accession | KEGG term | Gene name | Modified cysteine site(s) | Functional description |
|---|---|---|---|---|
| Q8DWB9 | Butanoate metabolism | adhE | 156, 317, 826, 868 | Aldehyde-alcohol dehydrogenase |
| Q59934 | | pfl | 382 | Formate acetyltransferase |
| Q8CWY9 | Nitrogen metabolism | gltA | 190, 285, 1012 | Glutamate synthase (Large subunit) |
| Q8CWY8 | | gltB | 108, 112, 309, 429 | Glutamate synthase (Small subunit) |
| Q8DVU9 | | glnA | 209 | Glutamine synthetase type 1 glutamate—ammonia ligase |
| Q8DUL2 | | SMU_913 | 148, 322 | Glutamate dehydrogenase |
| Q8DRR2 | Purine metabolism | guaB | 310 | Inosine-5'-monophosphate dehydrogenase |
| Q8DS46 | | rpoB | 519, 884 | DNA-directed RNA polymerase subunit beta |
| Q8DU81 | | guaA | 158, 197, 252 | GMP synthase [glutamine-hydrolyzing] |
| Q8DRY2 | | nrdD | 625 | Putative anaerobic ribonucleoside-triphosphate reductase |
| P26283 | Pyruvate metabolism | ldh | 74 | L-lactate dehydrogenase |
| Q8DSN9 | | accC | 95, 338 | Putative acetyl-CoA carboxylase biotin carboxylase subunit |
| Q8DTF8 | | pckA | 236, 324 | Phosphoenolpyruvate carboxykinase |
| Q59939 | Carbon metabolism | citZ | 13 | Citrate synthase |
| P72484 | | tpiA | 127, 128, 188 | Triosephosphate isomerase |
| Q8DVV3 | | gapC | 153, 157 | Glyceraldehyde-3-phosphate dehydrogenase |
| Q8DUP4 | Alanine, aspartate and glutamate metabolism | carA (pyrA) | 76 | Carbamoyl-phosphate synthase small chain |
| Q8DTY0 | | glmS | 2, 493 | Glutamine—fructose-6-phosphate aminotransferase [isomerizing] |
| Q8DWG0 | Pentose phosphate pathway | fbaA | 235 | Fructose-1,6-biphosphate aldolase |
| Q59931 | | gapN | 360 | NADP-dependent glyceraldehyde-3-phosphate dehydrogenase |
| Q8DTX6 | | pfk | 35 | ATP-dependent 6-phosphofructokinase |
| Q8DSW8 | Aminoacyl-tRNA biosynthesis | metS | 31 | Methionine—tRNA ligase |
| Q8DSL1 | | valS (syv) | 501 | Valine—tRNA ligase |
| Q8DWG1 | Pyrimidine metabolism | pyrG | 438 | CTP synthase |
| Q8DTV1 | | pyrF | 155 | Orotidine 5'-phosphate decarboxylase |
| Q8DTV0 | | pyrDB | 24 | Dihydroorotate dehydrogenase |
| Q8DS05 | Glycolysis/Gluconeogenesis | ptsG | 498 | Putative PTS system, glucose-specific IIABC component |
| Q8DV96 | Peptidoglycan biosynthesis | murF | 449 | UDP-N-acetylmuramoyl-tripeptide—D-alanyl-D-alanine ligase |
| Q8DST2 | | murE | 99 | UDP-N-acetylmuramoyl-L-alanyl-D-glutamate—L-lysine ligase |
| Q8DVE3 | | murD | 260 | UDP-N-acetylmuramoylalanine—D-glutamate ligase |

**Table 2. Differentially modified modification sites (modified proteins) summary.**

| Compare group | Regulated type | fold change >1.2 | fold change >1.3 | fold change >1.5 | fold change >2 |
|---|---|---|---|---|---|
| T/WT | up-regulated | 50 (41) | 13 (13) | 5 (5) | 1 (1) |
| | down-regulated | 21 (14) | 9 (6) | 4 (4) | 1 (1) |
| AB/WT | up-regulated | 29 (24) | 14 (12) | 4 (4) | 0 (0) |
| | down-regulated | 95 (63) | 62 (42) | 26 (21) | 4 (4) |

*Filtered with threshold value of modification fold change.

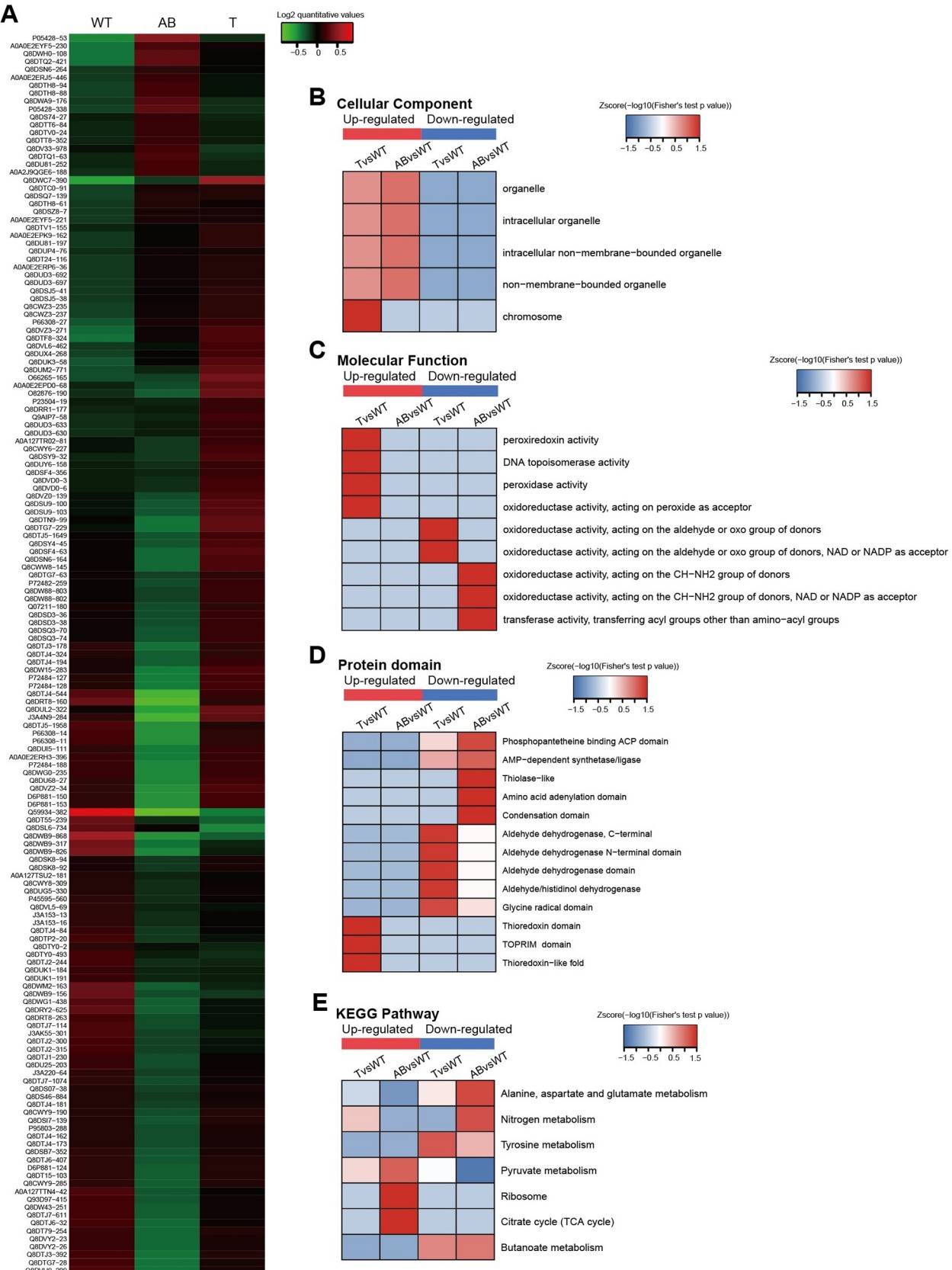

**Fig 4. S-glutathionylation of proteins is differentially modified after deletion of *gshAB* and *gsT*.** (A) A heatmap was used to visualize the significant differentially modified sites. log2(quantitative value) > 0 means that the modification level was upregulated and log2(quantitative value) < 0 indicates that the level was downregulated. (B) GO enrichment-based cluster analysis of the cellular components of differentially modified proteins. (C) GO enrichment-based luster analysis of the molecular functions of differentially modified proteins. (D) Domain enrichment-based cluster analysis of differentially modified proteins. (E) KEGG enrichment-based cluster analysis of the pathways of differentially modified proteins. The Z-score represents the–$\log_{10}$(Fisher's test *P*-value in enrichment analysis).

The domain description and GO, and KEGG annotations are shown in **S7 Table**. Then, we performed the clustering of differentially modified proteins based on the GO, KEGG, and domain enrichment analyses. In the "cellular component" category (**Fig 4B**), the differentially modified proteins after the deletion of *gsT* (group "T vs WT") were primarily enriched in the chromosome, including DNA gyrase subunit B and DNA topoisomerase 1, which are important for DNA replication and transcription. The functions of the differentially modified proteins were mainly enriched in various oxidoreductase activities (**Fig 4C**), indicating that S-glutathionylation modification alterations caused by the deletion of *gshAB* and *gshT* might affect the redox status of important molecules, such as NAD. In addition, the group "T vs WT" showed that some upregulated proteins were related to the peroxidase activity, suggesting that the deletion of *gsT* may cause the S-glutathionylation of some peroxidases to resist oxidative stress. Protein domain clustering also revealed the role of S-glutathionylation after *gsT* and *gshAB* deletion (**Fig 4D**). Interestingly, the thioredoxin domain and thioredoxin-like fold were significantly enriched after *gsT* knockout, which makes the peroxidase-like thiol peroxidase (tpx) resistant to oxidative stress. KEGG clustering indicated differentially modified proteins participated in central metabolic pathways, including TCA cycle and pyruvate metabolism pathway (**Fig 4E**). These results demonstrated that the deletion of *gsT* and *gshAB* would affect the modification of many proteins and their functions, subsequently affecting multiple metabolic pathways of *S. mutans*.

## S-glutathionylation of Tlp is important for the competitive ability of *S. mutans*

From the proteomics study, we concluded that protein S-glutathionylation might have crucial effects on the oxidative stress resistance as some protein containing thioredoxin-like fold could be S-glutathionylated at their active Cys sites. Previous studies have demonstrated that oxidative stress resistance ability is closely related to the competitive ability and virulence of *S. mutans* [15, 38]. Thus, we wanted to explore the role of S-glutathionylation in the regulation of the oxidative stress resistance ability and virulence of *S. mutans*. We identified a thioredoxin-like fold containing protein through domain annotation (Q8DSJ5, **S7 Table**, named as Tlp in this study) coded by *SMU_1788c*, which contained modified Cys sites at its thioredoxin catalytic motif ($C^{38}PYC^{41}$) and C100 (**Fig 5A** and **5B**, **S1 Table**). Moreover, the structural prediction revealed a similar structure with the classical Trx structure, which contains a highly conserved fold that is composed of five β-strands surrounded by four α-helices and a CXXC catalytic motif.

Interestingly, C38 and C41 of Tlp were significantly differentially modified after the deletion of *gsT*, whereas the function of this protein has never been studied. Homology comparisons with other common pathogens indicated that all these three cystine sites are substantially conserved (**Fig 5C**). To validate the proteomics data, we cloned and purified Tlp and its site-specific mutants for the C38, C41, and C100 fused with C-terminal 6 × His-tag through the pET21a protein expression system. Subsequent anti-glutathionylation western blotting indicated decreasing signals of S-glutathionylation on point mutant proteins (**Fig 5D** lanes 1–3). We also constructed a site-specific mutation for the protein Bta (AAL00156.1, bacteriocin

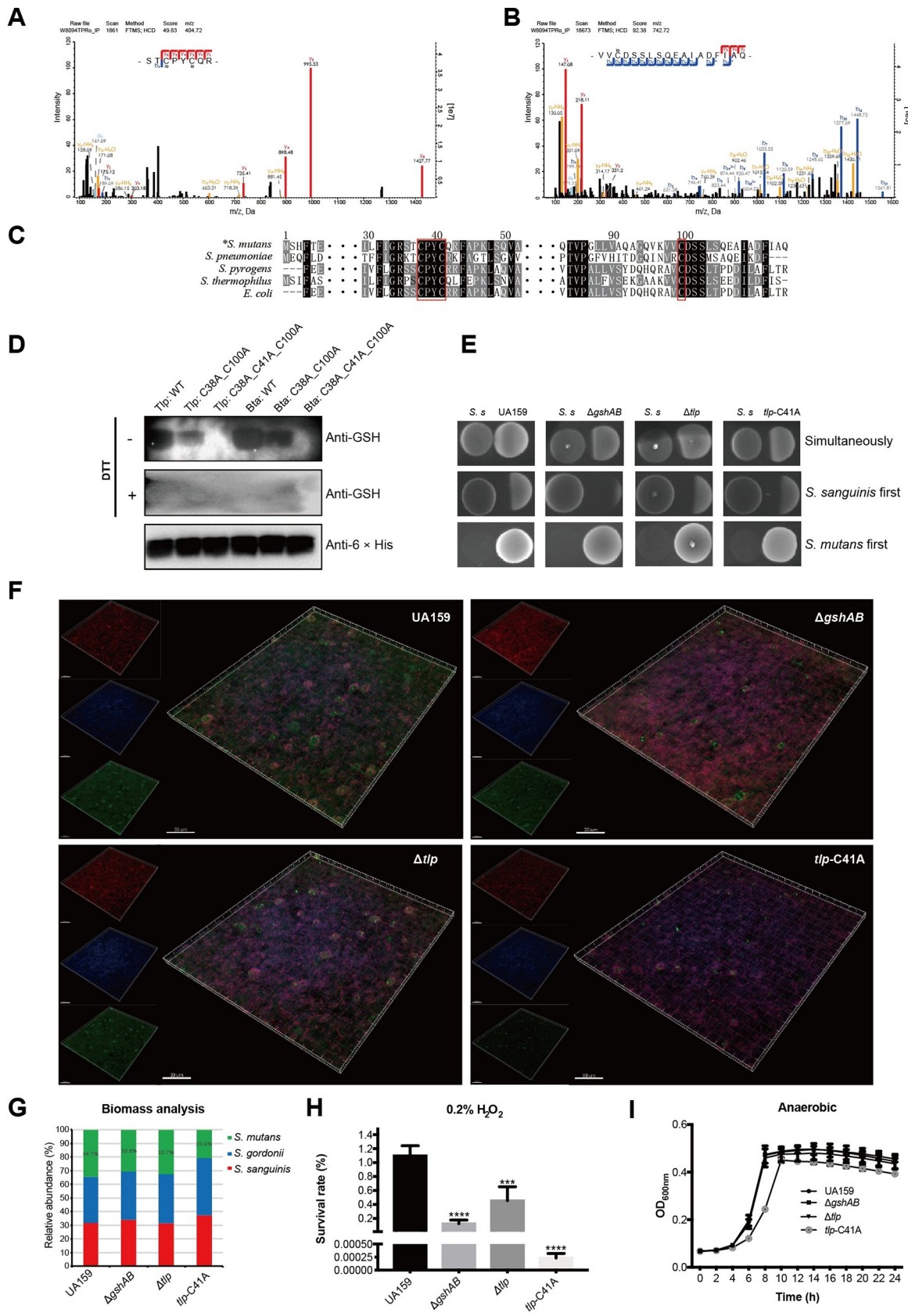

**Fig 5. S-glutathionylation of Tlp is important for the competitive ability of *S. mutans*.** (A) and (B) MS/MS spectral identification of S-glutathionylated peptides on the C38, C41, and C100 of Tlp. (C) Alignment of partial Tlp sequence homologs from other common bacterial species, including *Streptococcus pneumoniae* (WP_000434650.1), *Streptococcus pyrogens* (SQE57836.1), *Streptococcus thermophilus* (WP_011227571.1), and *Escherichia coli* (OWC47010.1). (D) Anti-glutathionylation western blotting for the target proteins and their point mutants, lanes 1–3: S. mutans *Tlp* and the mutants; lanes 4–6: *S. pneumoniae* Bta. (E) Competition between *S. mutans* and S. sanguinis on half-strength BHI plates. (F) Fluorescence image of the tri-species biofilm. *S. gordonii* (blue), *S. sanguinis* (red) and *S. mutans* (green) were labeled with species-specific FISH probes. (G) Bar graph showing the biomass ratio of each species created with Image Pro Plus 6.0. Results were averaged from five randomly selected positions of each sample. (H) Sensitivity of *S. mutans* UA 159, *gshAB* and *tlp*-C41A mutant strains to 0.2% $H_2O_2$. The results were averaged from three independent experiments and are presented as mean ± standard deviation (SD), *, P < 0.05; **, P < 0.01; ***, P < 0.001, by one-way ANOVA with Dunnett's multiple-comparison test. (I) Growth curves of *S. mutans* UA 159, *gshAB* and *tlp*-C41A mutant strains within 24 h with intervals of 2 h under anaerobic conditions. The results were averaged from three independent experiments and are presented as mean ± SD. We did not draw the error bars that were shorter than the height of the symbol.

transport accessory protein) of *Streptococcus pneumonia* R6, which is highly homologous to the Tlp we studied, and also contains three Cys residues at its thioredoxin catalytic motif ($C^{38}PYC^{41}$) and C100 (**Fig 5C** row 2). Anti-glutathionylation western blotting also showed decreasing signals on mutant proteins (**Fig 5D** lanes 4–6), suggesting that these Cys residues could indeed be S-glutathionylated. These results further verified the accuracy and reliability of our proteomics study, and suggest that S-glutathionylation of Tlp and its homolog proteins might be conserved and functional for other bacterial strains.

To better understand the effects of S-glutathionylation on the biological function of Tlp, we constructed a site-specific mutagenesis mutant for the C41 site by replacing Cys with alanine (A). It is worth pointing out that we could not replace the C38, probably because the replacement of C38 with Ala is lethal. We first investigate the effects of the Tlp-C41A mutant on the competition with *S. sanguinis* ATCC10556 on the solid plate. As shown in **Fig 5E**, the growth rates of the *ΔgshAB*, *Δtlp* and *tlp*-C41A mutant strains were inhibited by *S. sanguinis* compared to WT UA159 when *S. mutans* and *S. sanguinis* were inoculated simultaneously. When *S. sanguinis* was inoculated first and allowed to grow overnight, the growth of UA159 was also being inhibited but less severe than *ΔgshAB*, *Δtlp* and *tlp*-C41A mutant strains. However, all these four strains could inhibit the growth of *S. sanguinis* almost completely when *S. mutans* was inoculated first. We then used FISH to investigate the microbial composition and interaction among *S. mutans*, *S. sanguinis* and *S. gordonii* within a tri-species biofilm. Compared to that of the UA159, the abundance of *ΔgshAB* mutant strains was significantly decreased, and the abundance of the *tlp*-C41A mutant strain decreased most significantly. In contrast, the decrease in the abundance of the *Δtlp* strain was less severe than those of the other two mutant strains (**Fig 5F**). Biomass analysis of these three species showed that the abundance of *S. mutans* decreased from 35% to approximately 21% when the C41 of Tlp was replaced (**Fig 5G**). These interspecies competition assays indicated that the competitive ability of *S. mutans* was significantly decreased after the S-glutathionylation site lost.

$H_2O_2$ from *S. sanguinis* and *S. gordonii* is the main substance that causes *S. mutans* growth inhibition [15]. We performed an $H_2O_2$ sensitivity assay, and the results showed that the oxidative resistance abilities of the *ΔgshAB* and *Δtlp* mutant strains was significantly lower than that of UA159 (**Fig 5H**). Surprisingly, the survival rate of the *tlp*-C41A mutant strain was approximately 4000-fold lower than that of UA159, indicating that the S-glutathionylation of Tlp C41 was crucial for the oxidative resistance of *S. mutans*. To investigate whether the *ΔgshAB*, *Δtlp* and *tlp*-C41A mutants affect the growth of *S. mutans*, we performed a series of *in vitro* growth assays under anaerobic and aerobic conditions. As the growth curves show (**Fig 5I and S3 Fig**), there were no significant differences among the growth rates of *ΔgshAB* and *Δtlp* mutant strains and UA159. In contrast, the growth rate and the final bacterial amount of the *tlp*-C41A mutant strain were significantly lower than those of UA159. Interestingly, the *tlp*

null mutant did not affect the above phenotypes as C41A mutant, which might be because the overall deletion of *tlp* induced other antioxidant systems of *S. mutans* to resist oxidative stress. However, the substitution of C41 causes Tlp to be continuously activated and affects downstream signaling pathways, which leads to the growth inhibition of *S. mutans* and increased susceptibility to oxidative stress. These results suggest that the S-glutathionylation of Tlp C41 is vital for *S. mutans* to resist oxidative stress and compete with other oral streptococci.

## S-glutathionylation of Tlp is important for the cariogenecity of *Streptococcus mutans* in a rat caries model

The *tlp*-C41A mutant inhibited the growth and survival rates of *S. mutans* within the cariogenic biofilm, which cause decreased the virulence of *S. mutans*. We compared the cariogenecity potentials of UA159, *ΔgshAB* and *tlp*-C41A mutant strains in a rat model of dental caries [39]. The evaluation of caries by Keyes' scoring method revealed that the *ΔgshAB* and *tlp*-C41A mutants significantly reduced the severity of carious lesions on all molar surfaces compared to that in UA159 (**Table 3, Fig 6A and 6B**). As shown in Fig 5A and 5B, the carious lesions of group UA159 on buccal surfaces were much more severe than in the other groups. However, the differences between the UA159 group and the mutant groups on sulcal were not significant as on the buccal surfaces, which may result from the differences in oxygen concentrations and the exposed areas of bacteria to oxygen among different molar surfaces. The buccal surface is smooth, whereas the sulcal surface is bumpy, so the tiny grooves may be covered by food debris or other symbiotic microorganisms, creating hypoxic niches for the oxygen-sensitive strains to survive and even erode the enamel [40]. For the proximal surface, there were no dentinal extensive (Ex) lesions in either the gshAB or tlp-C41A mutant group. In conclusion, both the GshAB and the S-glutathionylation of Tlp are important for the virulence of *S. mutans* in vivo, indicating that GSH and protein S-glutathionylation play key roles in caries development.

## Discussion

Collectively, our study indicated that protein S-glutathionylation is widespread in the cariogenic bacteria *S. mutans* and is present in the active domains of various proteins. These proteins participate in many important metabolic processes and may affect the DNA replication, protein synthesis, energy metabolism, antibiotic resistance and virulence factor production. The deletion of *gshAB* and *gsT* affected the S-glutathionylation of many proteins in *S. mutans*, and these molecular functions are mainly enriched in various oxidoreductase and peroxidase activities, indicating that these modifications may have important effects on oxidative stress

**Table 3.  *gshAB* and *tlp*-C41A mutant reduced carcinogenicity of *S. mutans* in a gnotobiotic rat model.**

| Infecting strain | Mean caries scores (n = 5) | | | | | | | | | | |
| --- | --- | --- | --- | --- | --- | --- | --- | --- | --- | --- | --- |
| | Buccal | | | | Sulcal | | | | Proximal | | |
| | E | Ds | Dm | Dx | E | Ds | Dm | Dx | E | Ds | Dm |
| UA159 | 34.2 ± 2.01 | 18.4 ± 1.72 | 12.2 ± 1.24 | 5.8 ± 1.02 | 13.2 ± 0.97 | 10.2 ± 1.02 | 7 ± 0.71 | 4 ± 0.71 | 10.2 ± 1.24 | 4.8 ± 1.28 | 3 ± 0.63 |
| *ΔgshAB* | 13.6 ± 1.78[a] | 4.8 ± 0.58 [a] | 1.8 ± 0.58 [a] | 0.6 ± 0.40[b] | 14.2 ± 1.66 | 8.2 ± 1.77 | 5.6 ± 0.93 | 2.2 ± 0.80 | 6.8 ± 0.58 | 1 ± 0.32[d] | 0.0 |
| *tlp*-C41A | 10.6 ± 1.36[a] | 3.6 ± 0.81 [a] | 1.2 ± 0.37 [a] | 0.0 | 11 ± 1.00 | 4.4 ± 0.81[d] | 2.8 ± 0.66[c] | 0.2 ± 0.20[c] | 7.2 ± 1.02 | 1.6 ± 0.51[d] | 0.0 |

a: Significantly different from value for group UA159 (P < 0.0001)

b: Significantly different from value for group UA159 (P < 0.001)

c: Significantly different from value for group UA159 (P < 0.01)

d: Significantly different from value for group UA159 (P < 0.05)

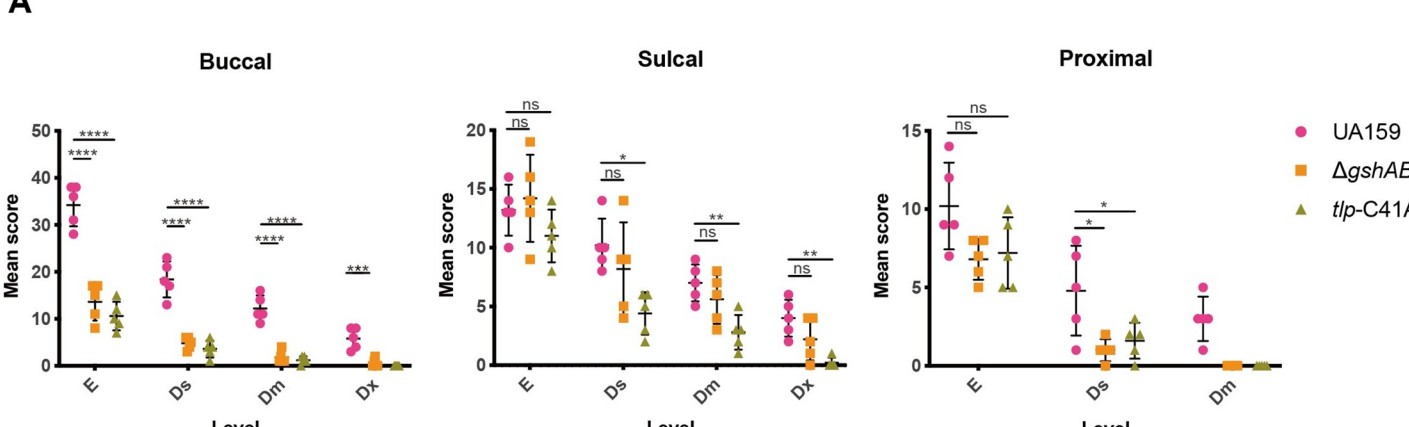

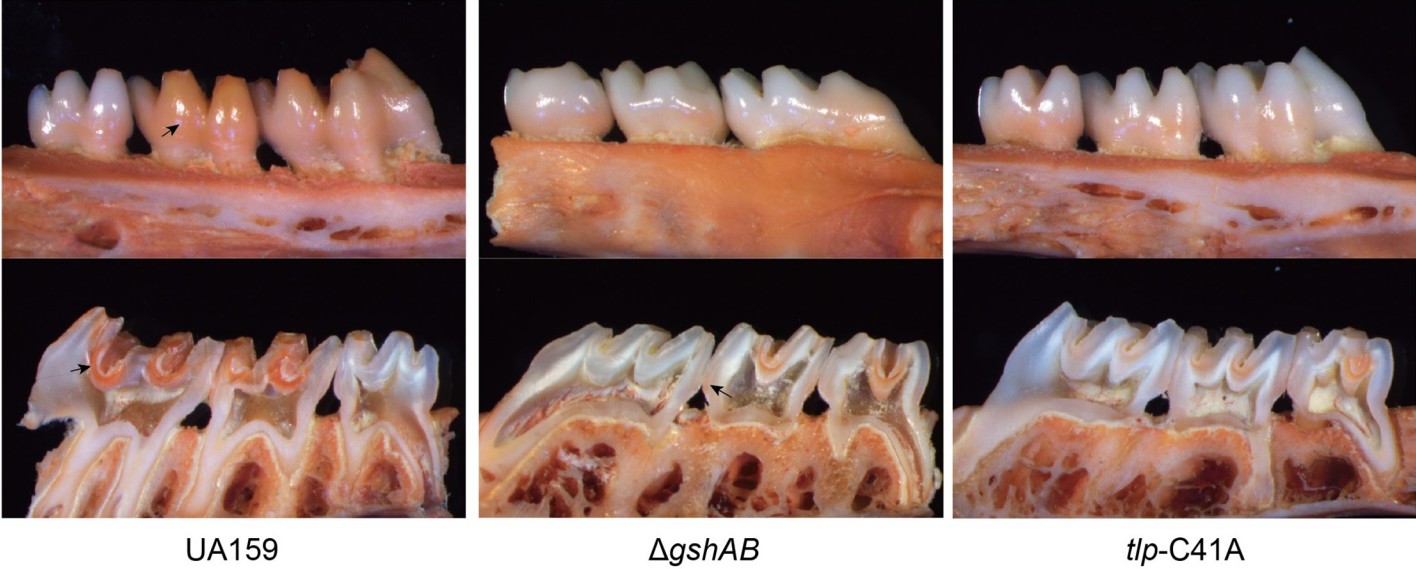

UA159 ΔgshAB *tlp*-C41A

**Fig 6. S-glutathionylation of Tlp is important for the cariogenecity of Streptococcus mutans in a rat caries model.** (A) Statistical chart of the caries scores in Table 3. Each plot represents the caries score of each rat (n = 5). Error bars denote the SD, *, P < 0.05; **, P < 0.01; ***, P < 0.001; ****, P < 0.0001, by one-way ANOVA with Dunnett's multiple-comparison test. E, enamel; Ds, dentinal slight; Dm, dentinal moderate; Dx, dentinal extensive. (B) Maxillaries of the model rats, The upper-panel shows the buccal view, the lower-panel shows the sulcal view, and arrows indicate the representative carious lesions.

resistance and redox signal regulation. One of the differentially modified proteins, Tlp, contains two modified Cys sites on its thioredoxin catalytic motif, and its function in the oxidative stress resistance has not yet been studied. Our study proved that S-glutathionylation of the Tlp C41 plays a vital role in oxidative stress resistance and interspecies competition in *S. mutans*. A subsequent rat caries model showed that GSH and Tlp glutathionylation significantly affected the cariogenecity of *S. mutans* significantly. However, there are still many proteins involved in vital metabolic pathways that would be valuable to study. The PPI network also revealed some central proteins, which will provide significant resources for further in-depth studies in the future to elucidate the role of S-glutathionylation in systemic signaling cascades.

Cysteine is a highly sensitive amino acid that can be oxidized by various ROS and subjected to multifarious reversible PTMs, including S-nitrosylation, S-glutathionylation and S-palmitoylation, collectively referred to as S-thiolations [41]. Among these S-thiolations, S-glutathionylation is the major modification that has crucial biological and functional significance in cellular signal transduction and diverse metabolic pathways [1, 34]. With the growing biological significance of S-glutathionylation, many biochemical methods have been developed to investigate the S-glutathionylated peptides and the development of proteomics technologies make it possible to quantify the S-glutathionylation on a large scale. Proteomic-based indirect identification can be achieved in a few steps: free thiol blocking [42], the selective reduction of S-glutathionylated cysteines [43], newly reduced thiols labeling [44], resin-assisted or biotin avidin-based enrichment [45], and HPLC-MS/MS identification and quantification. There are also some direct methods for the proteomics identification of the modified peptides, such as using anti-GSH antibodies to pull down the S-glutathionylated proteins. However, the low sensitivities of these antibodies limit the detection potential [46]. Recently, an isotopically labeled "clickable" GSH method was developed to quantify protein S-glutathionylation. In this method, a mutant of glutathione synthetase (GS), capable of coupling γGlu-Cys to azido-Ala in place of Gly was transfected into cells. It efficiently produced the azido-GSH which is then glutathionylated on proteins. The subsequent click reaction allows for enrichment and quantification [47]. However, as the "clickable" GSH contains an azide handle, it may interfere with enzyme-mediated (de)glutathionylation. In this study, we used N-ethylmaleimide (NEM) and iodoacetamide (IAM) to block free thiols. The selective reduction of S-glutathionylated cysteine with Grx reduction mixture (GSSG/GR/NADPH), and a specific reaction with the thiol (-SH) group isobaric mass label iodoTMT was used to modify the modified thiol, then resin conjugated with anti-TMT antibodies was used to enrich labeled peptides. The specificity and sensitivity of iodoTMT in labeling the reduced sulfhydryl group, enrichment efficiency of anti-TMT resin, and accuracy and precision of the MS methods have been measured in previous studies [23, 48, 49]. Combining these results with our quality control data proved our workflow has high specificity, accuracy and precision in the identification and quantification of the modified peptides. However, our approach also has several limitations. For example, the S-glutathionylation state could be easily interfered with by the redox state of the buffer solution and environmental cues, which might cause the thiol group of the cysteine residue to be converted to a reduced state and blocked by the NEM or IAM. In addition, as the labeling step is approved for undenatured and undigested proteins, some thiols may be blocked by the spatial structure, and would not be labeled. These two points are also the common limitations of the above approaches, which will cause the number of identified sites to be undercounted. It is worth noting that with the development of deep learning and artificial intelligence technology, computers will likely become important tools to help us predict PTM sites. A novel computing framework DeepGSH (http://deepgsh.cancerbio.info/), which is based on depth learning and particle swarm optimization algorithms, was developed for the prediction of S-glutathionylation [50]. However, this system can only be applied to Homo sapiens and Mus musculus at present, and its accuracy needs to be verified.

In some physiological processes and stress conditions, Cys residues can be oxidized to sulfenic acid or disulfide bonds, and the oxidized Cys residues can be reduced to the thiol state by various oxidoreductases, including thioredoxins (Trxs) and glutaredoxins (Grxs). Trxs from all kinds of organisms share a highly conserved fold that is composed of five β-strands surrounded by four α-helices, and a CXXC catalytic motif [51]. Under physiological conditions, the N-terminal Cys of this motif is mostly present as a thiolate, which attacks an oxidized thiol and forms a mixed disulfide bond between Trx and its substrate. Then the C-terminal Cys is deprotonated and engages in the nucleophilic attack of the disulfide bond, which releases an

oxidized Trx and a reduced substrate with a reduced thiol group. Thioredoxin reductase reduces the oxidized CXXC motif through an NADPH-dependent reaction [51, 52]. Grxs share some structural similarity to Trxs; most Grxs contain a CXXC catalytic motif, while the rest have a CXXS motif [53, 54]. Similar to Trxs, the CXXC motif of the Grxs reacts with oxidized proteins and is oxidized when the target disulfide is reduced. GSH then reduces oxidized Grxs with a two-step reaction, resulting in the release of reduced Grx and an oxidized GSH (GSSG). Trxs are ubiquitous in bacteria, whereas Grxs is absent in some specific bacteria, such as *Bacteroides fragilis* and *Lactobacillus casei* [55, 56]. Streptococci often contain both Grxs and Trxs, but do not possess catalase to catabolize the $H_2O_2$ directly; therefore, the antioxidant function of streptococci is mainly dependent on the thiol-dependent proteins [57]. For *S. mutans*, the major thiol-dependent antioxidant system is the Trx system. Marco's study revealed the function of three thioredoxins (TrxA, TrxH1, and TrxL) in *S. mutans* UA159 [58]. The Tlp we focused on this study is a 115-amino acids protein that contains a Trx-like fold in its region 4–114, indicating that it may function as a Trx in *S. mutans*, Our results revealed its crucial role in the oxidative stress resistance and interspecies competition. Moreover, it affected the growth of *S. mutans* after the replacement of its S-glutathionylated Cys residue at the catalytic motif.

We hypothesize that S-glutathionylation in other bacteria may also affect virulence and pathogenicity by regulating multiple protein functions and physiological processes. A comprehensive identification of S-glutathionylation would give us a complete understanding of the functions of this PTM in bacteria, which would further aid in pathogen treatment and disease prevention.

## Supporting information

**S1 Fig. Mass detection and identification of S-glutathionylated peptides and proteins in *S. mutans* UA159.** (A) Mass error distribution of all identified S-glutathionylated peptides in *S. mutans* UA159; (B) S-glutathionylated peptide length distribution in *S. mutans* UA159; (C) The number of S-glutathionylated sites identified per protein in *S. mutans* UA159.
(TIF)

**S2 Fig. COG functional classification chart of the S-glutathionylated proteins in *S. mutans* UA159.** The numbers of modified proteins are labeled at the top of per category, below the histogram chart are the description of the categories.
(TIF)

**S3 Fig. Growth curves of *S. mutans* UA 159, gshAB and tlp-C41A mutant strains within 24 h with intervals of 2 h under aerobic condition.** The results were averaged from 3 independent experiment and are presented as men ± SD. We didn't draw the error bars those are shorter than the height of the symbol.
(TIF)

**S1 Table. Information of modified peptides identified.**
(XLSX)

**S2 Table. Localize predication of the modified proteins.**
(XLSX)

**S3 Table. GO enrichment of the modified proteins.**
(XLSX)

**S4 Table. COG functional classification of the modified proteins.**
(XLSX)

**S5 Table. Protein domain enrichment of the modified sites.**
(XLSX)

**S6 Table. KEGG pathway enrichment of the modified proteins.**
(XLSX)

**S7 Table. Annotations combine of the modified proteins.**
(XLSX)

**S8 Table. Primers used in this study.**
(XLSX)

**S1 Data. Excel spreading containing, in separate sheets, the underlying numerical data and statistical analysis for figure panels 2A, 2B, 2C, 2D, 3A, 3B, 4A, 4B, 4C, 4D, 4E, 5G, 5H, 5I, 6A, S1A, S1B, S1C, S2, S3.**
(XLSX)

## Author Contributions

**Conceptualization:** Zhengyi Li, Xuedong Zhou.

**Data curation:** Zhengyi Li, Xin Xu, Xian Peng, Xuedong Zhou.

**Formal analysis:** Zhengyi Li, Chenzi Zhang, Xin Xu, Xian Peng, Xuedong Zhou.

**Funding acquisition:** Xian Peng, Xuedong Zhou.

**Investigation:** Zhengyi Li, Chenzi Zhang, Cheng Li, Xian Peng.

**Methodology:** Zhengyi Li, Cheng Li, Xin Xu, Xian Peng.

**Project administration:** Zhengyi Li, Chenzi Zhang, Xian Peng.

**Resources:** Zhengyi Li, Jiajia Zhou, Xian Peng.

**Software:** Zhengyi Li, Cheng Li, Jiajia Zhou.

**Supervision:** Xin Xu, Xian Peng, Xuedong Zhou.

**Validation:** Zhengyi Li, Chenzi Zhang, Cheng Li, Jiajia Zhou, Xian Peng, Xuedong Zhou.

**Visualization:** Zhengyi Li, Chenzi Zhang, Cheng Li, Jiajia Zhou.

**Writing – original draft:** Zhengyi Li.

**Writing – review & editing:** Zhengyi Li, Xin Xu, Xian Peng, Xuedong Zhou.

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
