## [Decision Letter · Decision Letter 0]

24 Feb 2020

Dear Dr. Peng,

Thank you very much for submitting your manuscript "S-glutathionylation proteome profiling reveals the crucial role of a thioredoxin-like protein in competitiveness and cariogenecity of Streptococcus mutans" for consideration at PLOS Pathogens. As with all papers reviewed by the journal, your manuscript was reviewed by members of the editorial board and by several independent reviewers. In light of the reviews (below this email), we would like to invite the resubmission of a significantly-revised version that takes into account the reviewers' comments.

The reviewers were generally favorable about this submission, but had a number of questions and suggestions for improvement, as detailed below, which will need to be addressed.

In addition, there were underlying questions posed by at least one reviewer are whether there another glutathionylation system, does TlpA have homologues in other bacteria, if so what is known about them? In revising this submission, it would be good to consider addressing these issues, since doing so would help clarify the work. In addition, one reviewer was especially concerned about the writing, and the need to improve clarity and grammar.

We cannot make any decision about publication until we have seen the revised manuscript and your response to the reviewers' comments. Your revised manuscript is also likely to be sent to reviewers for further evaluation.

Sincerely,

Paul M Sullam

Associate Editor

PLOS Pathogens

Michael Wessels

Section Editor

PLOS Pathogens

Kasturi Haldar

Editor-in-Chief

PLOS Pathogens

orcid.org/0000-0001-5065-158X

Michael Malim

Editor-in-Chief

PLOS Pathogens

orcid.org/0000-0002-7699-2064

The reviewers were generally favorable about this submission, but had a number of questions and suggestions for improvement, as detailed below, which will need to be addressed.

In addition, there were underlying questions posed by at least one reviewer are whether there another glutathionylation system, does TlpA have homologues in other bacteria, if so what is known about them? In revising this submission, it would be good to consider addressing these issues, since doing so would help clarify the work. In addition, one reviewer was especially concerned about the writing, and the need to improve clarity and grammar.

Reviewer's Responses to Questions

**Part I - Summary**

Reviewer #1: S-glutathionylation is an important post-translational modification that has been extensively studied in eukaryotes. Briefly, it protects proteins from oxidative stress and can alter protein function. In eukaryotes, S-glutathionylation is widespread and therefore impacts many physiological processes, whether this is the same for G+bacteria was unknown – although two studies with L. monocytogenes and S. pneumonaie suggest it will be the case.

The authors examined aspects of gluthationylation in S. mutans. This is relevant as this cariogenic bacterium must deal with considerable oxidative stress due to its competition with other H2O2 producing streptococci. Importantly, S. mutans carries a dual-functional enzyme GshAB, which has both glutamylcysteine synthetase and glutathione synthetase functional domains. GshAB has been shown to be essential for the competitiveness and prevalence of S. mutans through detoxifying the H2O2 produced by S. sanguinis. However, the target proteins affected by GshAB are unknown. The submitted work addresses this lapse and sheds light on the importance of S-glutathionylation in G+ bacteria, with specific detail provided as to how S. mutans deals with oxidative stress.

Overall the work is well done. However, and as described below, expansion and clarification of the text is required. So is improvement of the statistical analyses. In addition, the manuscript would be improved by addition (if possible) of a TlpA deletion mutant.

If addressed, the submitted manuscript substantially advances our understanding of the impact of glutathionylation in G+ bacteria and provides an elegent example of its importance.

Reviewer #2: This study identifies potential S-glutathionylated proteins in S. mutans proteome. It also identifies the role of a novel Trx-like protein in a caries model. One of the major concern with this study is that data is descriptive still authors make bold conclusions. Protein-s-glutathionylation is important for survival; however, not critical. Also, control experiments with one or two purified proteins showing the residues identified can indeed be S-glutathionylated is missing. Overall, the work is interesting; however, presenting data only as functional categories or as their subcellular location in the bacteria downplays the significance of many interesting proteins.

Reviewer #3: In this manuscript Li and colleagues investigate the posttranslational modification of cysteine residues by glutathione (GSH) in Streptococcus mutans. GSH is not found in most Gram positive organisms but some can synthesize it and/or import it, and evidence suggests that S. mutans is at least able to import GSH. From earlier work it appears that GSH is an important component of S. mutans physiology and loss of gshAB has previously been shown to impact S. mutans competitiveness with other oral streptococci. No proteome-wide study, as reported here, has been carried out before. Several mass spectrometry methodologies now exist to identify Cys-redox modifications, and the authors use an interesting methodology to do so. After showing proteome-wide modifications of Cys the authors then move on to study a protein, Tlp, whose Cys modification is important for growth, competition and virulence. The study is interesting and contributes to an obviously important but perhaps still poorly understood facet of Gram positive/streptococci physiology. However, aspects of the experimental design, or at least the explanation of it, are flawed.

**Part II – Major Issues: Key Experiments Required for Acceptance**

Reviewer #1: 1. The authors delete gshAB (which is responsible for synthesizing GSH) and Gst (which conjugates GSH to proteins) and determine the number of proteins having STM by the same methods. Surprisingly, 337 modified sites on 224 proteins were quantifiable. In some instances levels of glutathionylation were increased for some of the STM proteins in these mutants. This indicates that there is another glutathionylation process occurring within the bacterium. This is very pertinent but was not discussed.

2. The homology of SMU_1788c to other known genes needs to be investigated thoroughly. There is a TlpA gene in S. pneumoniae that has been demonstrated to be important for survival against oxidative stress. Is this the same gene?

3. The authors should include results with a tlpA deletion mutant. In turn describe how tlp-c141A compares to the null mutant? It’s possible that by altering aa141 the activity of TlpA is altered in fashion that makes in constantly activated – this would have a distinct phenotype than from its overall deletion. If a deletion mutant cannot be made this should be indicated.

4. There are broad concerns with regards to the statistics. These need to be addressed.

Reviewer #2: The authors approach to map all the S-glutahionylated proteins has been done nicely, and the data looks interesting. Nevertheless, the manuscript needs major revision. Some general concerns with the manuscript are as follows:

1. the grammar is incorrect at may places.

2. citations of similar articles on S-glutathiolation from other bacteria are missing. Also, citations are missing from the result section, specifically, where the authors discuss about various proteins in functional classification (page 14-15).

3. Growth condition mentioned in the materials and methods section is anaerobic. When authors talk about the growth profile of mutants, it is assumed that it was also carried out in the anaerobic chamber. If so, how do they expect that gshAB and gsT deletion would impact growth?

4. It is also not clear that why gshAB and gsT mutants were chosen, when the bacteria was grown anerobically and all the thiolation work was carried out using added glutathione using lysates. Table S1 lists the fold change in the mutants and wt, while it is not clear that simialr peptides/thiolation-sistes identified in both mutants and wt. Is it likely that the authors only picked common proteins/peptides identified in all three conditions. How do they justify excluding comparison with a redox-active agent with wt sample only rather than anaerobically grown wt and gshAB/T mutants for this work?

5. Group AB and T descrription missing from the method section. it only becomes clear in later part of result.

6. a lot of information from the result section about the function of various membrane proteins from the result section can go to discussion.

7. It will be nice if the authors show some tables showing the relevant proteins related to their functional categories as table, and this is quite common for omics data.

8. database search mentions search with UA159 database; however, Table S1 has protein accession numbers corresponding to GS5 too.

9. the data is descriptive except the tlp mutant, hence conclusions shoud be more suggestive rather than being close ended.

10.For the enrichment analysis of mutant mass spec data, it will be intersting to see a table in the text or a heat map of proteins rather tahn functional categories to highlight the important proteins affected in the mutants.

11. the rationale for calling Tlp a Trx like protein is missing. How does it compare with the known Trxs in terms of overall predicted structure, sequence similarity etc. Alo CXXC in Tlp does not have a positively charged amino acid next to cys and therefore thiol may not be a reactive one. The data related to Tlp with respect to caries is fine and it can be considered an important protein.

12. Discussion talks about the known methods to probe S-glutathionylation; however, relevant citation is missing. It would be more appropriate to compare the data with available data in literature.

13. Figures are not readable.

14. String analysis shows the connection between identified proteins. Hence the conclusion that all the glutathionylated proteins interact is wrong. They are a part of a network.

15. Figure legend for 5B is not clear.

16. A better title would be "S-glutathionylation proteome profiling reveals a crucial role of a thioredoxin-like protein in interspecies competition and cariogenecity of Streptococcus mutans".

Reviewer #3: 1) I make a minor comment below that is related to describing the methodology in greater detail, and some of my comments here might be in part related to this. Having pieced together the methodology/samples used I have some technical issues that I think the authors should consider. What is used to discriminate potential artifacts/contamination that arose during sample preparation? I do not think that a negative control is described (non-treated).

2) S. mutans cells are cultured in BHI in anaerobic conditions in this study. Is it not possible to perform the experiments in cells growing in various redox conditions? This would probably also help with studying changes in gshAB or gshT mutant backgrounds.

3) Were replicate experiments carried out? I cannot see this described in the methods/results.

**Part III – Minor Issues: Editorial and Data Presentation Modifications**

Reviewer #1: 1. Using high-specific labeling with high sensitivity mass spectrometry the investigators identified what is claimed to be the entire S-glutathionylation proteome of S. mutans UA159. A total of 357 cysteine S-glutathionylation sites on 239 proteins were identified. How does the fact that 11.7% of the S. mutans proteome can be glutathionylated compare to E. coli and Aeromonas hydrophila (if not bacteria then in eukaryotes)? Is the percentage of S. mutans PTM proteins low or high in comparison to these? How is this process the same or different between the species this has been examined?

2. It would be quite helpful if the investigators provided the distribution of proteins having 1-4 glutathionylated sites (do most have 1 or more than 1?).

3. In the discussion the authors provide detailed explanation for why the approach taken is best. Yet the do not discuss the limitations of their approach, i.e. any possible reasons that the methodology employed may be undercounting the number of identified proteins that are subject to S-glutathionlyation. This should be added.

Reviewer #2: Line 18: residues

Line 18: under conditions of oxidative...

Line 18: "at the condition of" can be changed to "during"

Line 25: iodoTMT...what does TMT stand for?

Line 27: Proteome profililing....(remove following)

Line 31: on interspecies competition...

Line 51: italicize S. mutans...follow this for whole manuscript.

Line 54: interspecies competition instead of competitiveness...

Line 61: addition of GSH, which causes an increase

Line 69 to 72: ref for S. pneumonie, L. monocytogenes gsh work...

Line 89: hostile to what?

Line 132: "The primers used in this listed are listed in Table S8." instead of performed.....

Line 196: is it MASCOT or Masquant?

Line 203: iodo TMT 6 plex var statement not clear.

competition assay: ref for 50% BHI plate?

Line 278: "reduced" to be changed as "reduction"

Line 284: segments or amino acids?

Line 289: indispensable to be replaced by important...gshAB mutants are not lethal

Replace UNIPROT ids with SMU no. it is easy for teh readers to follow.

Line 310-311: which adherence assay? or what adhesin? citations missing for function of all the proteins discussed in page 15.

Page 18: ref for role of gshAB and gsT in S. mutans

Reviewer #3: 1) It would be helpful if the method used to examine Cys-redox modifications was explained better/earlier in the manuscript. It took me a significant amount of reading (from other papers) to elucidate how it works and then determine caveats. A diagram, as part of a figure, could be especially helpful with this.

2) Line 468 “tlp-C41A mutant strain were significantly decreased compared with UA159” – is this a significant contributing factor to the decreased virulence in vivo?

3) Which strain of UA159 is used? Recently it was found that certain strains of UA159 have a truncation in perR (Smu.593) that might impact how it responds to H2O2 etc.

4) Is it known if gshAB is functional in UA159? I cannot find a careful study of GSH synthesis in S. mutans and a paper by Sherrill and Fahey suggests that synthesis is not possible (PMCID: PMC107044).

5) The quality of written English could be improved throughout.

PLOS authors have the option to publish the peer review history of their article (what does this mean?). If published, this will include your full peer review and any attached files.

Reviewer #1: Yes: Carlos Javier Orihuela

Reviewer #2: Yes: surabhi mshra

Reviewer #3: No
---

## [Editor Report · Decision Letter 1]

22 Jun 2020

Dear Dr. Peng,

Thank you very much for submitting your manuscript "S-glutathionylation proteome profiling reveals a crucial role of a thioredoxin-like protein in interspecies competition and cariogenecity of Streptococcus mutans" for consideration at PLOS Pathogens. As with all papers reviewed by the journal, your manuscript was reviewed by members of the editorial board and by several independent reviewers. The reviewers appreciated the attention to an important topic. Based on the reviews, we are likely to accept this manuscript for publication, providing that you modify the manuscript according to the review recommendations.

While the scientific issues have been addressed, the manuscript itself remains difficult to read, due to errors in grammar and syntax.  In some sections, these problems significantly impair the clarity of the work, We would recommend getting additional editorial assistance to address this issue.

Sincerely,

Paul M Sullam

Associate Editor

PLOS Pathogens

Michael Wessels

Section Editor

PLOS Pathogens

Kasturi Haldar

Editor-in-Chief

PLOS Pathogens

orcid.org/0000-0001-5065-158X

Michael Malim

Editor-in-Chief

PLOS Pathogens

orcid.org/0000-0002-7699-2064
---

## [Editor Report · Decision Letter 2]

1 Jul 2020

Dear Dr. Peng,

We are pleased to inform you that your manuscript 'S-glutathionylation proteome profiling reveals a crucial role of a thioredoxin-like protein in interspecies competition and cariogenecity of Streptococcus mutans' has been provisionally accepted for publication in PLOS Pathogens.

Best regards,

Paul M Sullam

Associate Editor

PLOS Pathogens

Michael Wessels

Section Editor

PLOS Pathogens

Kasturi Haldar

Editor-in-Chief

PLOS Pathogens

orcid.org/0000-0001-5065-158X

Michael Malim

Editor-in-Chief

PLOS Pathogens

orcid.org/0000-0002-7699-2064
---

## [Editor Report · Acceptance letter]

17 Jul 2020

Dear Dr. Peng,

We are delighted to inform you that your manuscript, "S-glutathionylation proteome profiling reveals a crucial role of a thioredoxin-like protein in interspecies competition and cariogenecity of Streptococcus mutans," has been formally accepted for publication in PLOS Pathogens.

Best regards,

Kasturi Haldar

Editor-in-Chief

PLOS Pathogens

orcid.org/0000-0001-5065-158X

Michael Malim

Editor-in-Chief

PLOS Pathogens

orcid.org/0000-0002-7699-2064